# Particle Shape Recognition with Interferometric Particle Imaging Using a Convolutional Neural Network in Polar Coordinates

Alexis Abad, Alexandre Fahy, Quentin Frodello, Barbara Delestre, Mohamed Talbi and Marc Brunel *

Unité Mixte de Recherche du Centre National de la Recherche Scientifique 6614 CORIA,
University of Rouen Normandy, Avenue de l'Université, BP 12, 76801 Saint-Etienne du Rouvray, France;
abada@coria.fr (A.A.)
* Correspondence: brunel@coria.fr; Tel.: +33-232953733

**Abstract:** A convolutional neural network (CNN) was used to identify the morphology of rough particles from their interferometric images. The tested particles had the shapes of sticks, crosses, and dendrites as well as Y-like, L-like, and T-like shapes. A conversion of the interferometric images to polar coordinates enabled particle shape recognition despite the random orientations and random sizes of the particles. For the non-centrosymmetric particles (Y, L, and T), the CNN was not disturbed by the twin image problem, which would affect some classical reconstructions based on phase retrieval algorithms. A 100% recognition rate was obtained.

**Keywords:** interferometric particle imaging; speckle; convolutional neural network

## 1. Introduction

The characterization of particles in a flow is important in many domains from nuclear safety to meteorology, aircraft safety, pollutant emissions, sprays, and medicine. Due to the speed of the flow, which can exceed hundreds of meters per second (as in airborne applications or in wind tunnels, for example), single-shot measurement techniques are often required. Optical techniques based on the use of nanosecond pulsed lasers address this constraint and offer interesting possibilities. One attractive method is interferometric particle imaging (IPI), which is the object of this paper [1–14]. In this technique, particles are illuminated with a laser, and a CCD sensor records the interferometric image scattered by the particle (in an out-of-focus plane). It is an off-axis technique, and there is no reference beam incident on the sensor, contrarily to holography. First developed to characterize spherical droplets or bubbles [1–14], IPI was extended to the characterization of irregular rough particles [15–21]. In the case of droplets, two-wave interference fringes (parallel bright lines) are analyzed: the diameter of the droplet (or bubble) is proportional to the frequency of the fringes. In the case of rough particles, the interferometric image is a speckle pattern whose 2D Fourier transform is linked to the global shape of the particle [18]. The analysis of the speckle patterns can be limited to Fourier transforms to determine the size of a particle or to other algorithms, such as phase retrieval algorithms [22–24], to identify the shape. Deep learning offers a very interesting alternative [25–27], although it requires a very large amount of data for training, and such data are difficult to obtain. One main problem is that there is no exact model that is able to rigorously predict the interferometric image of an irregular rough particle of any shape and any orientation with respect to the imaging line. In addition, the number of possible shapes is infinite (from symmetric particles such as ice crystals to sand, ashes, or coal particles of any morphology without symmetry), while their sizes can vary from microns to millimeters, which completely affects the speckle patterns (from very large to very small specks of light, significantly modifying the image sampling conditions).

In this paper, we address the problem linked to the orientation of the rough particles under observation. This problem is important when realizing multi-view systems in order to perform an efficient 3D tomography of the particles. Convolutional neural networks (CNNs) are powerful tools in translation due to the convolution operator [25]. In our case, particles can rotate in 2D space, and a CNN cannot correctly classify these particles from their interferometric images for any orientation of the same particle. CNNs are not rotation-invariant algorithms [25]. To overcome this problem, we propose in this study to convert a particle's interferometric images from Cartesian coordinates to polar coordinates because a rotation in the Cartesian plane is a translation in polar coordinates [28,29]. We will show that an inception CNN architecture [30,31] is able to identify different families of particle shapes from their interferometric images converted to polar coordinates, regardless of the orientation and the size of the particles.

## 2. Experimental Setup and Image Conversion to Polar Coordinates

### 2.1. Programmable Rough Particles

In order to obtain a large amount of experimental data for CNN training, the use of "programmable particles" created with a digital micromirror device is very convenient. This setup was proposed in reference [32]. It has been shown to produce interferometric images that look like those that would be delivered by real rough particles of equivalent shapes with similar properties. More precisely, in interferometric particle imaging, the two-dimensional Fourier transform of the interferometric image of a particle can be assimilated to the two-dimensional autocorrelation of the contour of this particle. This relation was established theoretically using a simplified light scattering model and was validated experimentally with many rough particles (ice crystals, sand or coal particles, and ashes, for example). This property is perfectly reproduced with the DMD setup [32]. Experimentally, the orientation of an irregular particle in a flow is never perfectly known. As a consequence, the exact knowledge of the shape of a particle (from the angle of view of an imaging system) is always uncertain. This uncertainty makes the evaluation of the accuracy of the developed image processing techniques very approximative. Using the DMD setup, it is possible to program shapes that are perfectly known, and this limitation does not exist. The quality of the image analysis can be precisely evaluated. The DMD setup is currently used to develop and improve image processing algorithms in IPI. An incident laser beam is sent to a DMD (composed of millions of micromirrors). Only hundreds of the micromirrors are programmed "on-state" mirrors and reflect light in the direction of an out-of-focus imaging system. Each of these "on-state" micromirrors acts as a scattering asperity of a rough particle. The global shape of the particle is the envelope that includes all "on-state" micromirrors. The shape and the size of this envelope can be programmed "on the demand". In particular, the same particle can be programmed with different orientations to simulate any orientation of a rough particle in a flow. Technically, the DMD is illuminated by a HeNe laser (wavelength: 632.8 nm). The size of the whole DMD is 1.4 cm × 0.8 cm. The laser beam incident on the DMD is enlarged using a telescope in order to cover the whole surface of the DMD. The DMD is composed of 1920 × 1080 mirrors with a square geometry. The out-of-focus imaging system is composed of a Nikon objective (focus length = 80 mm), an extension ring, and a CCD sensor. This sensor records the interferometric out-of-focus images of the particles programmed on the DMD. It is a Thorlabs BC106N-VIS/M camera with 1360 × 1024 pixels and a pixel size of 6.45 μm × 6.45 μm. The parameters of the setup are those of reference [24].

To test the consequence of a conversion of the interferograms to polar coordinates, we tested a set of six different particle shapes. We generated rough sticks; crosses; dendrites; and L-like, T-like, and Y-like particles with the DMD. Examples of these shapes programmed on the DMD are reported in Figure 1a–f, respectively, for arbitrary orientations. Each cross-like, T-like, or L-like particle was composed of two perpendicular branches with similar lengths and widths. A Y-like particle was composed of three branches with the

same lengths and widths. Dendrite-like particles were composed of six branches with the same lengths and widths with ramifications at their extremities.

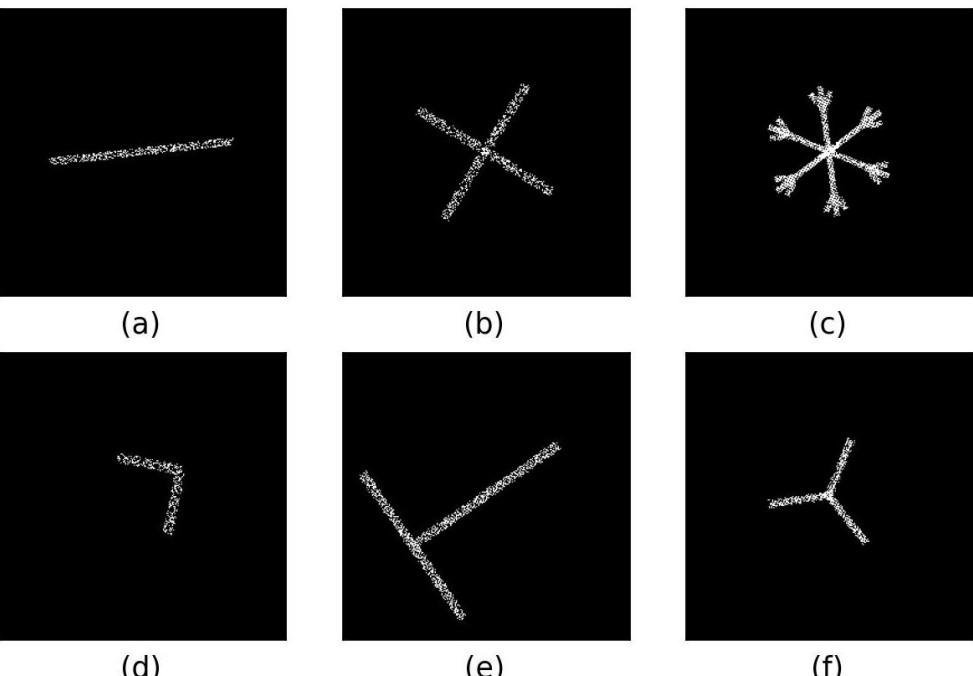

**Figure 1.** Typical rough stick (**a**), cross (**b**), dendrite (**c**), L-like (**d**), T-like (**e**), and Y-like (**f**) particles programmed on the DMD with arbitrary orientations. Each image is 2 mm × 2 mm.

In addition, the sizes of the programmed particles varied. For each particle's shape, the size (d) of the particle could vary in the range from 0.4 to 1.5 mm. Finally, the number of "on-state micromirrors" was proportional to the volume ($d^3$) of the programmed particle in order to be as near as possible to real cases. For example, a stick particle could be defined with 500 micromirrors, while a dendrite whose branches had the same length as the stick was defined with 1500 micromirrors. The interferometric experimental images were recorded on a CCD sensor composed of 1545 × 1164 pixels (with a pixel size of 6.45 μm), but the CNN only considered the center of these images: a central selection of 256 × 256 pixels.

### 2.2. Conversion of the Interferograms to Polar Coordinates

For all particles generated with the DMD setup, the interferometric pattern is a speckle pattern, similar to what would be generated by a real rough particle of equivalent shape. Unfortunately, for a given 2D shape, the orientations of the particles can vary: particles can suffer rotation. CNNs are not rotation-invariant algorithms, and a CNN will not correctly classify these particles. To overcome this problem, we converted the interferometric patterns from Cartesian to polar coordinates. A rotation in the Cartesian plane corresponds to a translation in polar coordinates. As CNNs are powerful tools in translation due to the convolution operator, the problem of the rotation of the particle is solved with this procedure [28,29]. Polar coordinates ($\rho, \theta$) are easily defined from Cartesian coordinates $(X, Y)$ using the classical relations $\rho = \sqrt{X^2 + Y^2}$ and $\theta = \tan^{-1}\left(\frac{Y}{X}\right)$. In a polar representation, the radius ($\rho$) is on the horizontal axis and the rotation angle ($\theta$) is on the vertical axis. Figure 2 illustrates the difference between these systems of coordinates in the case of a dendrite-like particle. Figure 2a shows the patterns of a dendrite in Cartesian coordinates, while Figure 2b shows its representation in polar coordinates. Figure 2c shows the dendrite after a rotation of 15 degrees, while Figure 2d shows it in polar coordinates. The

rotation in Cartesian coordinates corresponds to a translation in polar coordinates. Similar observations can be made in Figures 3 and 4 for L-like and T-like patterns, respectively.

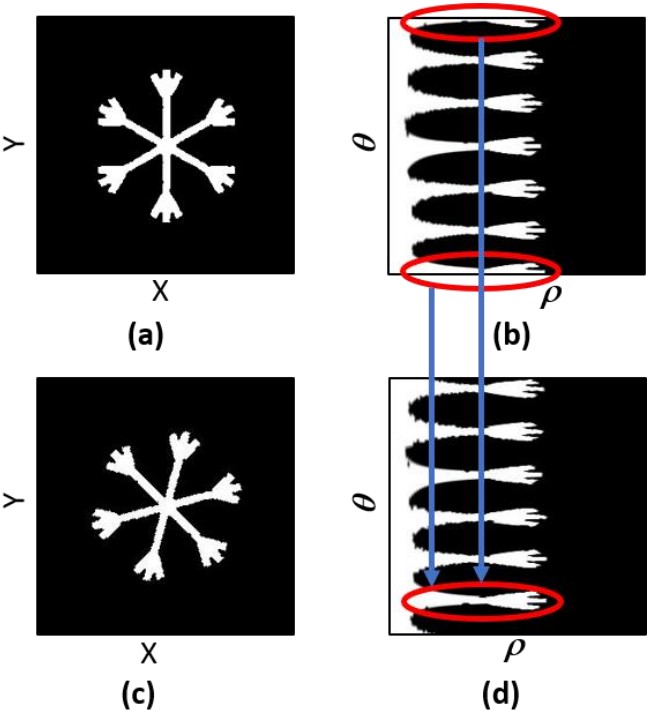

**Figure 2.** Dendrite-like particle represented in Cartesian coordinates (**a**,**c**) and polar coordinates (**b**,**d**) before (top) and after (bottom) a 15-degree rotation of the particle. The rotation in Cartesian coordinates is a translation in polar coordinates.

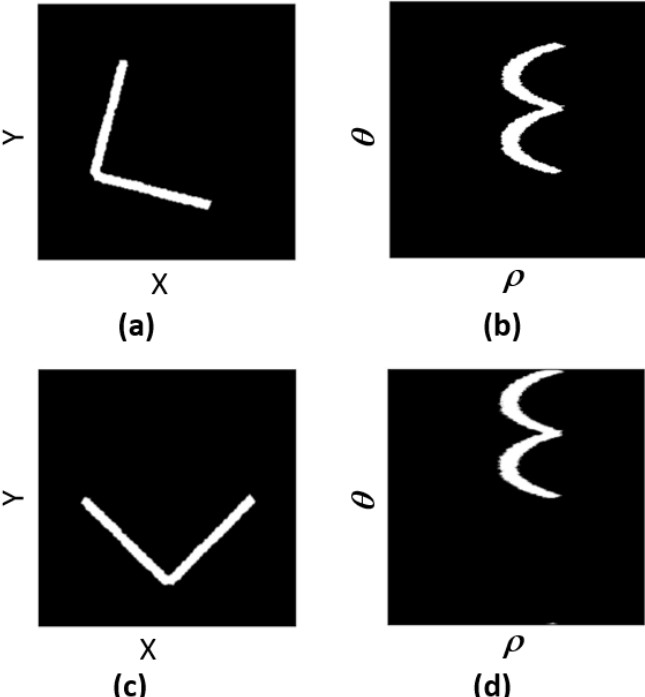

**Figure 3.** L-like particle represented in Cartesian coordinates (**a**,**c**) and polar coordinates (**b**,**d**) before (top) and after (bottom) a rotation of the particle.

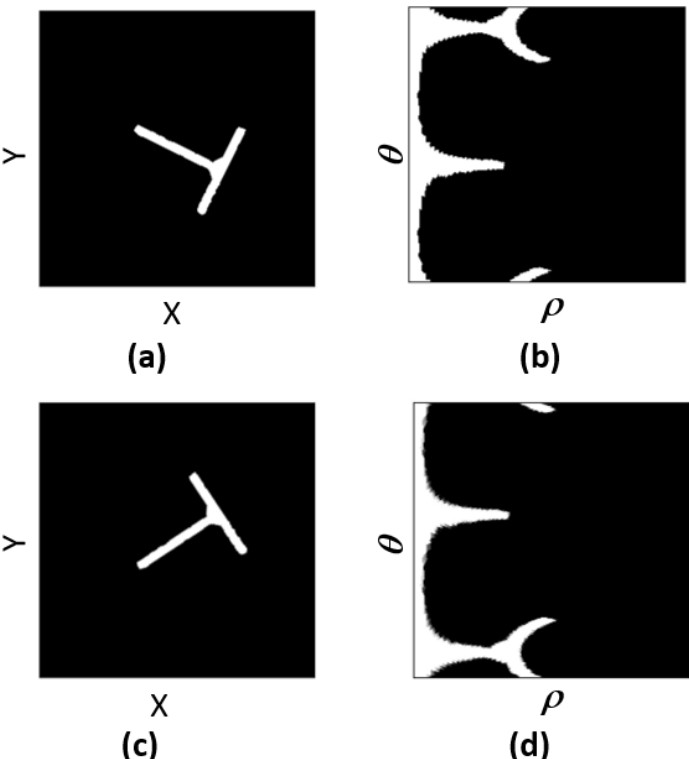

**Figure 4.** T-like particle represented in Cartesian coordinates (**a**,**c**) and polar coordinates (**b**,**d**) before (top) and after (bottom) a rotation of the particle.

This procedure was applied to the interferometric images of irregular rough particles. Figure 5a shows an interferometric image of a rough stick-like particle using the DMD setup in Cartesian coordinates. Figure 5b shows the same pattern in polar coordinates. A second stick-like particle was then programmed on the DMD. It had a similar size and shape, and it was composed of the same number of micromirrors, although the choice of the on-state micromirrors in the global envelope was random (and thus different). The main difference was that the stick had suffered a rotation of 60 degrees in the Cartesian coordinates. Figure 6a shows the interferometric image after this rotation of the particle in Cartesian coordinates, while Figure 6b shows the image after the rotation of the particle in polar coordinates. Figure 6a clearly shows the effect of the rotation of the particle. The main characteristics of the speckle pattern suffered a rotation of 60°. In polar coordinates, this rotation corresponded to a vertical translation of the main characteristics of the pattern. In Figures 5 and 6, we consider central selections of the interferometric patterns containing $256 \times 256$ pixels (camera pixel size of 6.45 µm). The axes are the same as in Figure 2: $(X, Y)$ in Figure 2a and $(\rho, \theta)$ in Figure 2b.

The role of the CNN is to recognize the characteristics of the patterns presented in polar coordinates for different shapes of rough particles and for different sizes of these particles.

### 2.3. CNN Architecture

Let us now describe the CNN that was tested. We recorded 500 speckle patterns for each particle shape (and thus a total of 3000 patterns) to create our database. We used 2400 patterns in grayscale for the training phase (actually, the central selections of $256 \times 256$ pixels in Cartesian coordinates were converted to polar coordinates, as previously explained), and each image was normalized between $-1$ and $1$. The choice of the dimensions of the selected part (i.e., $256 \times 256$) was a balance between computing time, memory capacity, and the number of speckle grains present in the selected part:

- Concerning the computing time and memory capacity, the network was trained on a GPU with 8 GB of memory capacity. With this capacity, the GPU could store the model, the batch of images, and the gradient calculation for each epoch.
- In interferometric particle imaging, particle sizing is possible when there is a sufficient number of fringes (for droplets) or speckle grains (for rough particles) in the image. The number of speckle grains in the selection is directly linked to the size of the particle and the geometry of the experimental setup (in particular, the defocus parameter). In the selected part of the image, we always had more than 30 bright spots. If this was not respected, the recognition of the smallest particles would not have been possible.

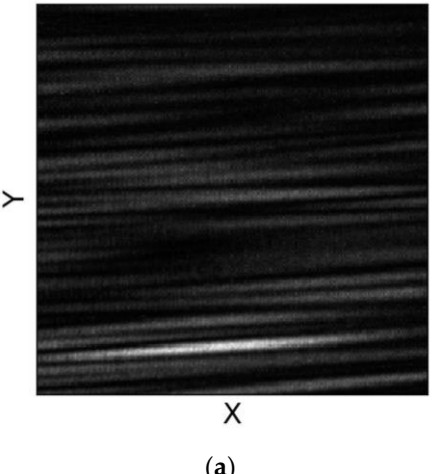 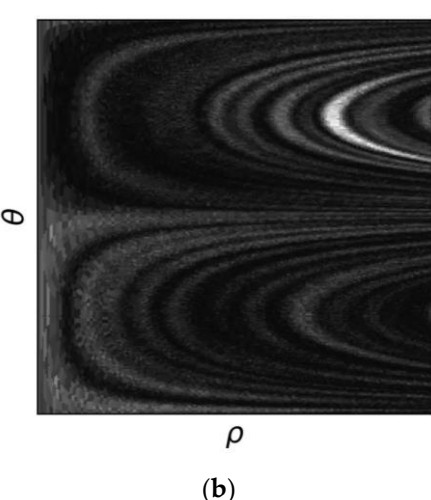

(**a**)                                                     (**b**)

**Figure 5.** Interferometric image of a stick-like rough particle in Cartesian (**a**) and polar coordinates (**b**). We present a central selection of the interferometric patterns containing $256 \times 256$ pixels (camera pixel size of 6.45 microns) in Cartesian coordinates and its conversion to polar coordinates.

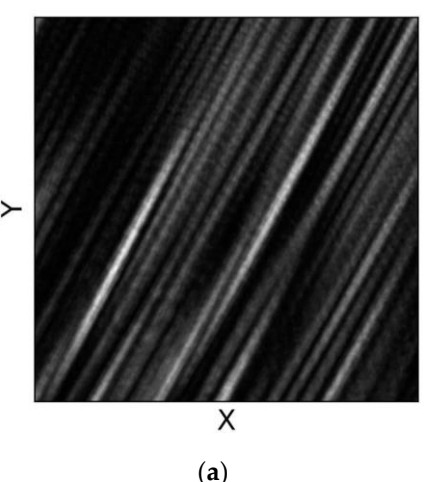 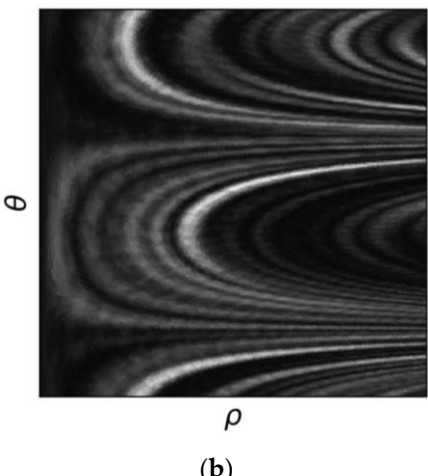

(**a**)                                                     (**b**)

**Figure 6.** Interferometric image of a similar stick-like rough particle after a 60° rotation in Cartesian (**a**) and polar coordinates (**b**). We present a central selection of the interferometric patterns containing $256 \times 256$ pixels (camera pixel size of 6.45 microns) in Cartesian coordinates and its conversion to polar coordinates.

In summary, the choice of the dimensions of the selected part was a compromise between the computation capabilities and the geometry of the setup. The selected part could of course be larger using more powerful computers.

The model shape of the CNN was based on two successive custom "inception" modules [30,31]. Figure 7 shows the CNN architecture. Each inception module was composed

of three branches: top, middle, and bottom. The top branch was composed of three successive convolution blocks with filter sizes of (1,1), (1,3), and (3,1); the middle branch was composed of five successive convolution blocks with filter sizes of (1,1), (1,3), (3,1), (1,3), and (3,1); and the bottom branch was composed of a single convolution block with a (1,1) filter. In each convolution layer, we set the number of output filters to 24. The network was trained on a computer with eight Tesla M10 GPUs with Pytorch 1.10.2. It took approximately 45 min for 50 epochs to occur. The learning rate was set to $5 \times 10^{-4}$ with an L2 regularization of $10^{-4}$, and the batch size was set to 32 images. The loss function that was used was the sparse categorical cross entropy, and we used the accuracy metric to measure learning evolution during training. We initially tested more simple architectures of CNNs using successive convolution, ReLU correction, and batch normalization, but the best results were obtained using inception modules, as described previously.

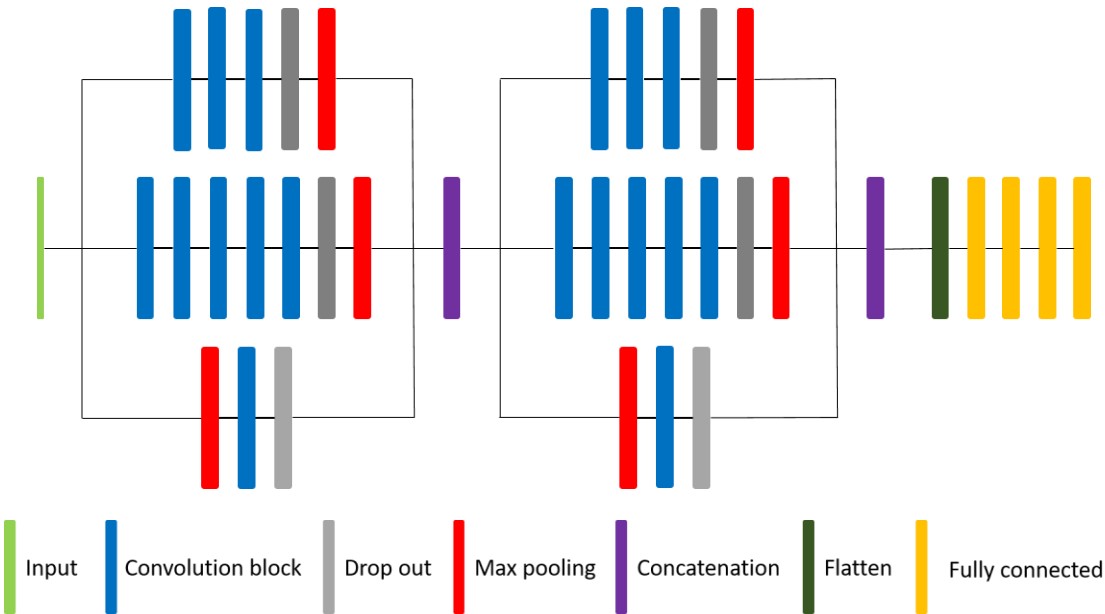

**Figure 7.** The CNN architecture to classify speckle patterns into 6 categories. From left to right, the image is fed to the CNN and gives a response. Gray rectangles represent dropout layers, red rectangles are maxpool layers, purple rectangles are concatenation layers, dark green rectangles are flatten layers, and yellow rectangles are fully connected layers. Blue rectangles are convolution blocks: one convolution layer followed by one batch normalization and one ReLU layer.

## 3. Results

### 3.1. Examples of Tested Data

Before presenting the results obtained with the CNN, let us visualize the kind of data the CNN has to compare and recognize. We can observe the differences induced by the shape of the particle in Figure 8. Figure 8a shows an interferometric image of a cross-like particle (four perpendicular branches); Figure 8b shows an image of a dendrite-like particle (six branches); and Figure 8c–e show interferometric images of non-centrosymmetric L-like, T-like, and Y-like particles, respectively. As previously stated, these patterns correspond to central selections of the interferometric images recorded with the CCD sensor containing $256 \times 256$ pixels. They were converted to polar coordinates. The axes are $(\rho, \theta)$ in all images in Figure 8. In addition, the size of the programmed particles is random in the range (400 µm, 1.2 mm). To illustrate this, Figure 8e shows an interferometric image of a Y-like particle (size of the particle: 660 µm), while Figure 8f shows an interferometric particle of similar shape whose branches are approximately half as long (size of the particle: 350 µm). As would be observed in Cartesian coordinates, the size of the speck of light is larger, and the number of specks of light is reduced when the particle is smaller.

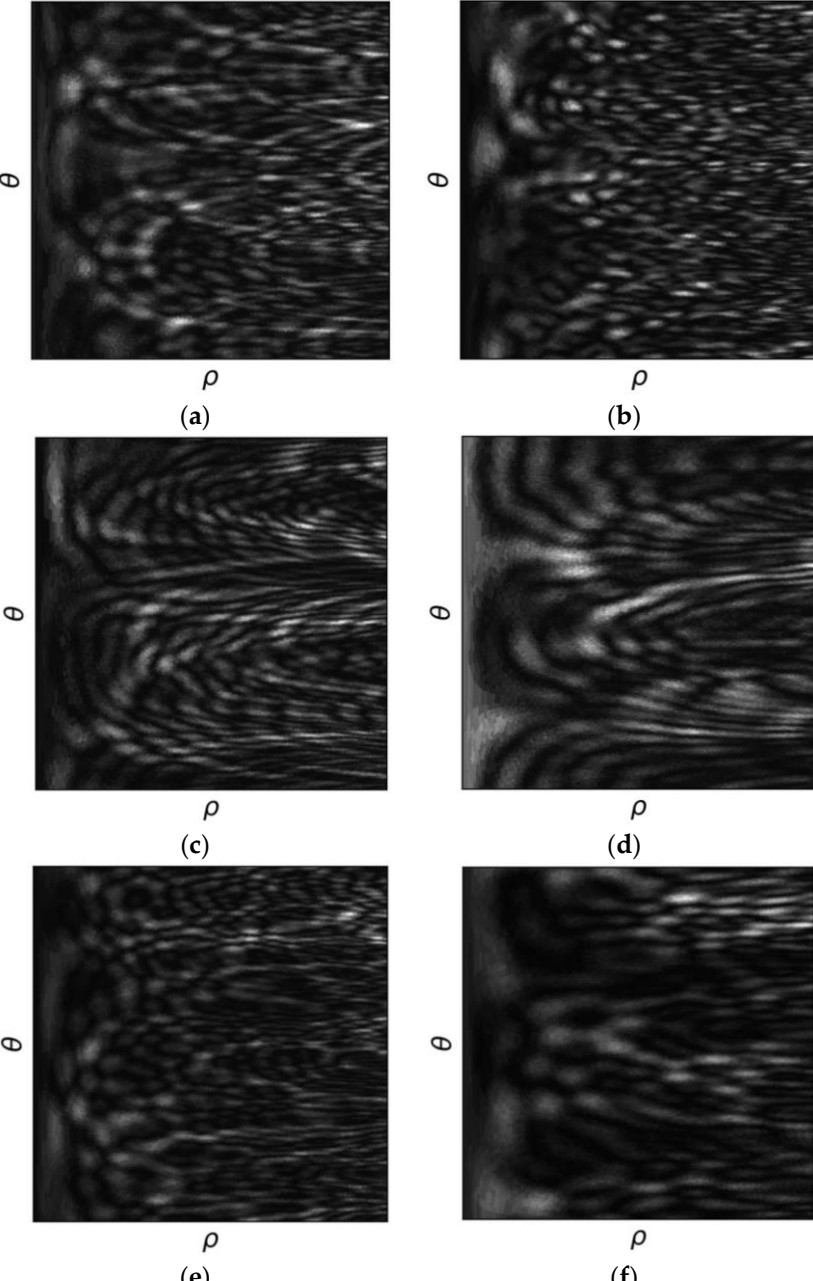

**Figure 8.** Interferometric images, after conversion to polar coordinates, of a cross-like particle with 4 branches (**a**), a dendrite-like particle with 6 branches (**b**), an L-like particle (**c**), a T-like particle (**d**), a Y-like particle (**e**), and a Y-like particle half the size of that in (**e**,**f**).

Figure 1 shows the six families of shapes that were programmed. However, during training and validation, there were no identical particles in our database. First, the size of each particle was random. Figure 9 shows a typical size histogram of the programmed particles in the case of the stick-like particles. Similar histograms could be plotted for the other families of particle shapes. In addition, the widths of the branches of each particle were random (in the range [30 μm, 80 μm]). The choice of the micromirrors programmed in the "on-state" in the global shape of the particle was random. Finally, the orientations of these particles in the transverse (X,Y) plane were also random. We used an ensemble of 3000 speckle patterns recorded with the DMD setup, corresponding to 3000 programmed particles. Due to the random programming of the particles, all particles were actually different.

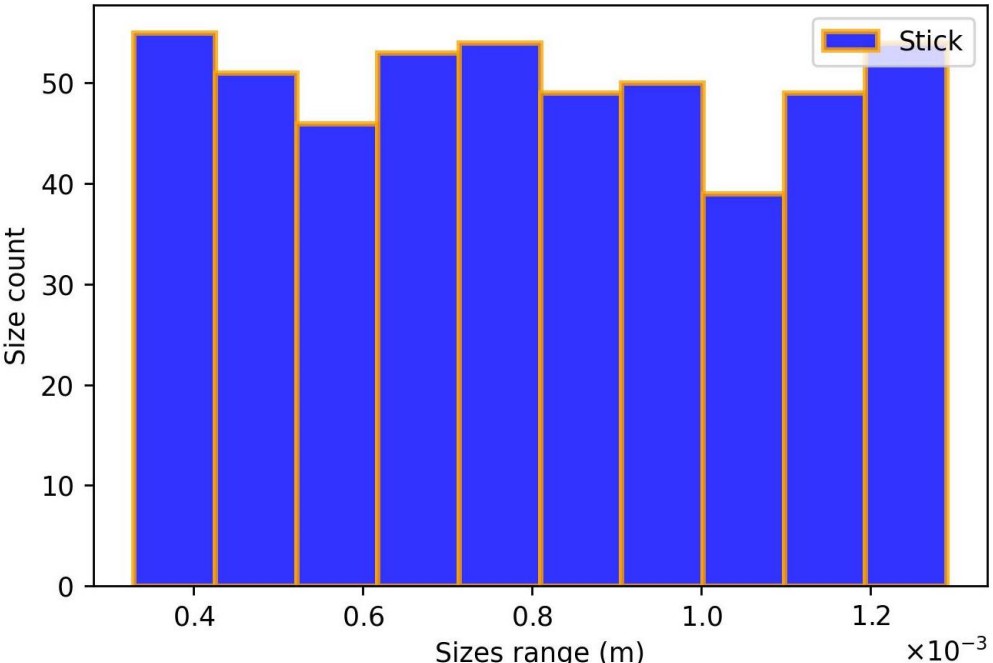

**Figure 9.** Histogram of the sizes of the stick-like particles programmed for training and validation.

The addition of the panel of non-centrosymmetric shapes was important because interferometric particle imaging can suffer from the twin image problem, as discussed using phase retrieval algorithms in a recent study [24]. All these patterns were experimental patterns obtained with the DMD setup.

### 3.2. Obtained Results

Figure 10 shows the model loss versus the number of epochs. From epoch 30, there was good convergence between the training data and validation data. For the last iteration, the validation loss was 0.0018. Using only 50 epochs, the model allowed the loss of three decades. Moreover, the training and validation curves converged with the same speed until the last epoch: we observed no under-fitting or over-fitting. For this reason, the model can easily generalize the classification problem. Figure 11 shows the accuracy metrics of the model. As the model learned, the accuracy increased, as expected, until it reached 100% accuracy over the validation dataset. This is a very satisfying result for image classification.

The test dataset was composed of 300 images. Figure 12 shows the confusion matrix of the whole test dataset. The model classified these images with 100% accuracy.

We could verify that the results were the same after a slight decentering of the programmed particles (within the limits of the DMD setup's capabilities, i.e., particles were decentered by 1 or 2 mm). This was favored by the fact that we analyzed the center of the interferometric images to avoid the defects of their borders. In the case of decentered particles, the coordinate change from Cartesian to polar coordinates had to be performed taking the center of each interferometric image as the origin.

To compare the efficiency of our neural network with the case where images were not converted to polar coordinates, Figure 13 shows the confusion matrix obtained with the same network when keeping the interferometric images in Cartesian coordinates. The neural network was trained in conditions similar to the previous ones. The validation results show that stick-like particles were still recognized without error, regardless of the sizes and orientations of the sticks. However, the error could exceed 30% of the tested cases for more complex objects such as cross-like or dendrite-like particles. The conversion to polar coordinates avoided the non-invariant rotational problem of convolution in the neural network to make the classification of interferometric images easier.

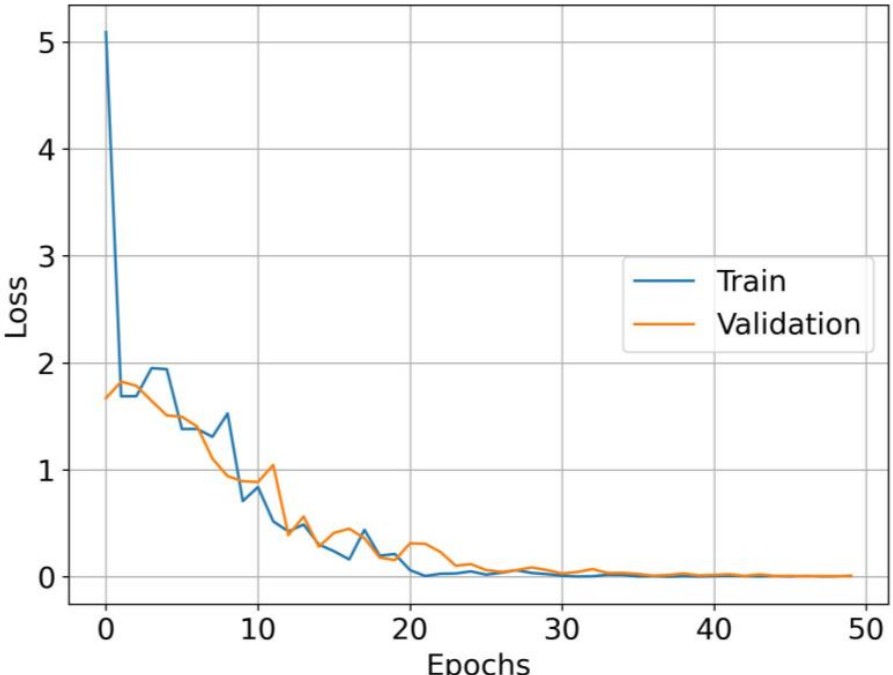

**Figure 10.** Loss versus number of epochs.

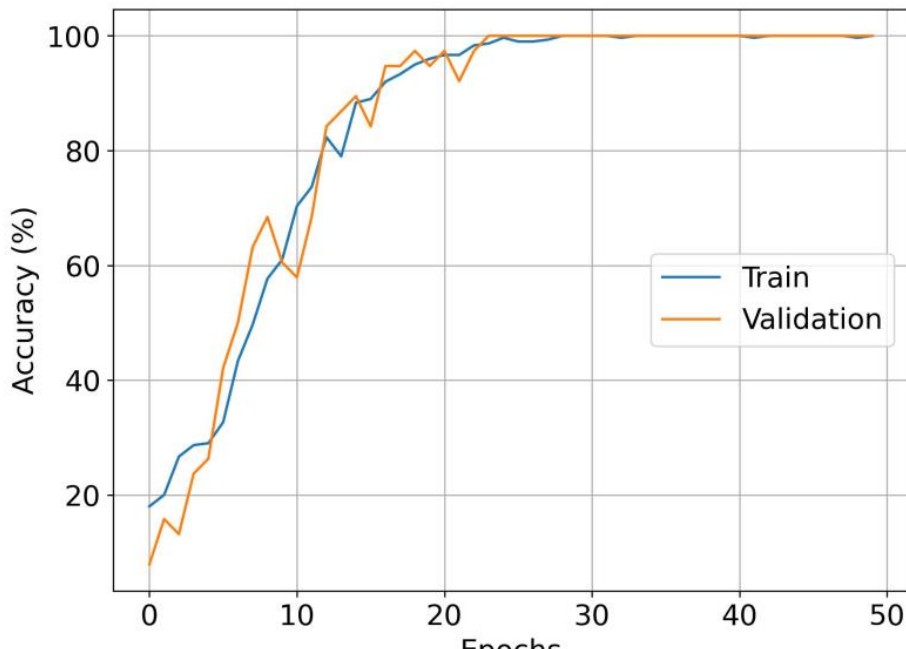

**Figure 11.** Accuracy versus number of epochs.

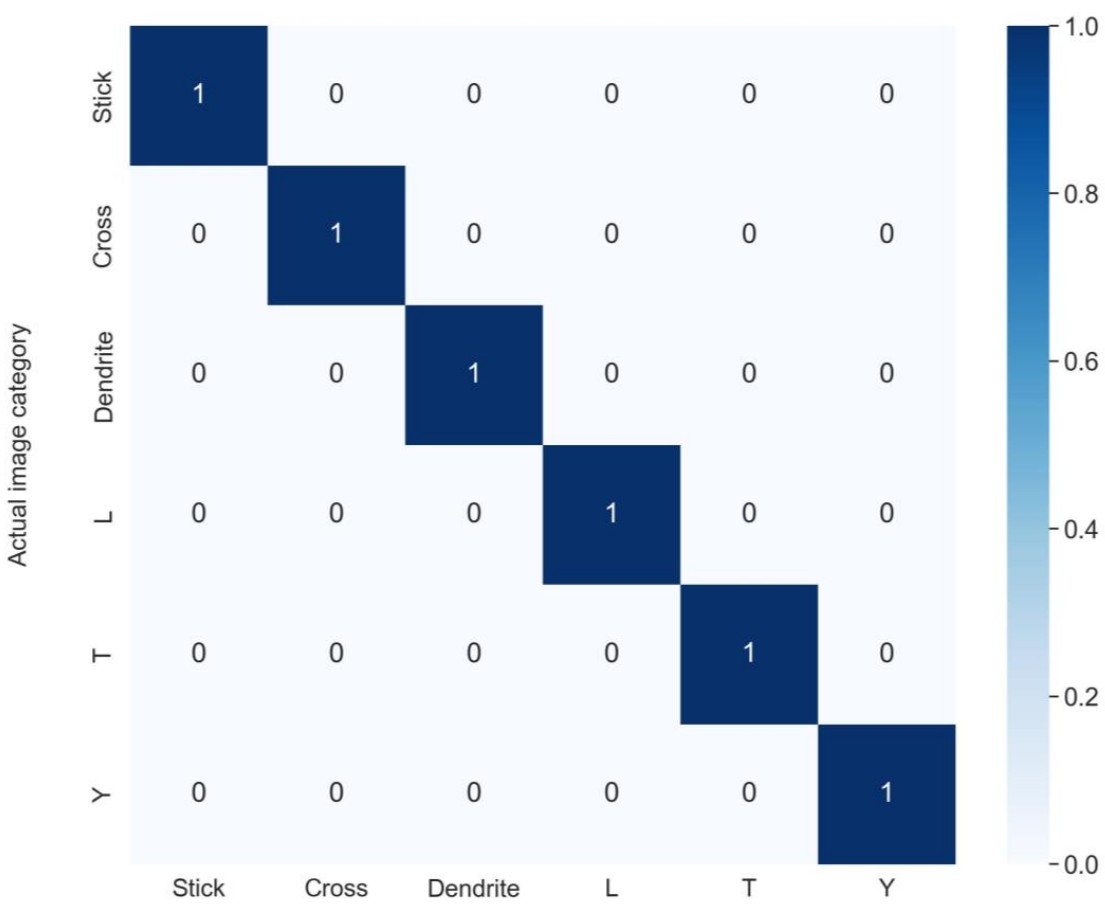

**Figure 12.** Confusion matrix of the test dataset. The vertical axis is the ground-truth image category, and the horizontal axis is the predicted image category.

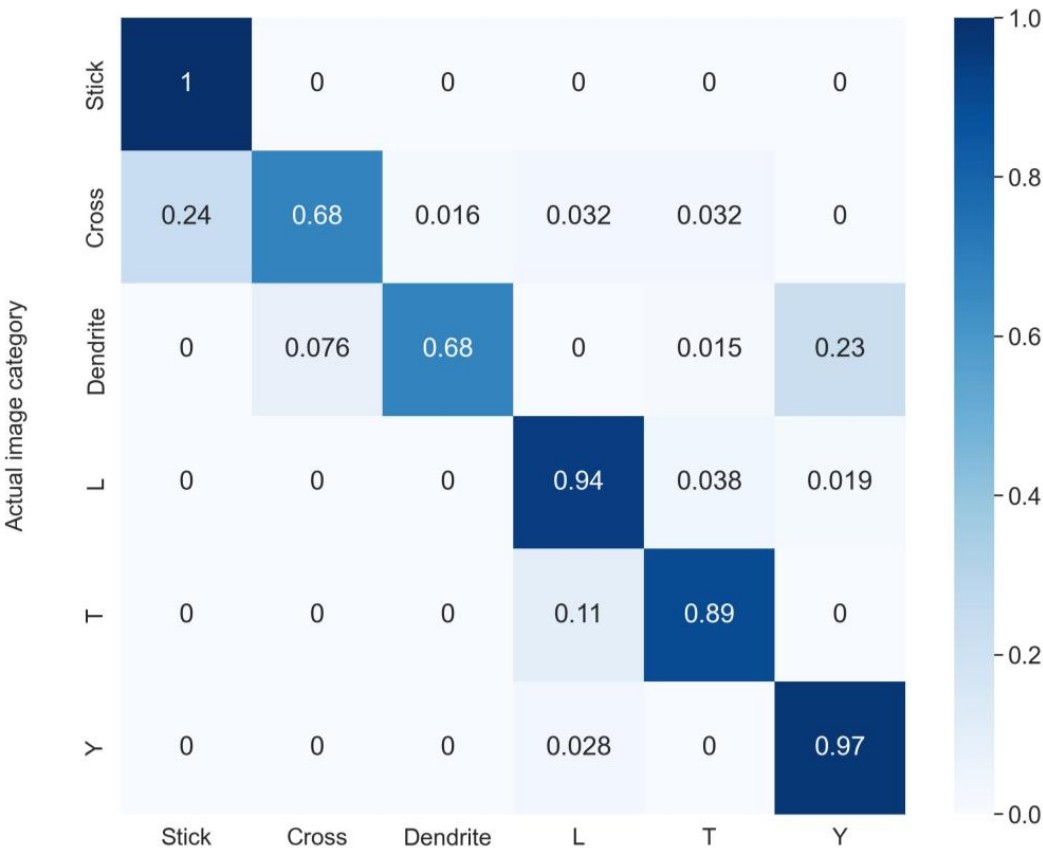

**Figure 13.** Confusion matrix with images in Cartesian coordinates. The vertical axis is the ground-truth image category, and the horizontal axis is the predicted image category.

## 4. Discussion

Over the last decade, computer vision has been pushed forward through the development of convolutional neural networks (CNNs). Interferometric particle imaging presents its own specificities, as this technique does not record real images of objects (i.e., the particles under observation); the interferometric images of rough particles are speckle patterns without direct links with the original objects. Nevertheless, it appears in this study that a CNN can be perfectly adapted to analyze these patterns and to allow particle shape classification. CNNs are limited by the lack of the ability to be spatially invariant to the input data. This could be very limiting in the case of IPI, where the sizes of the particles in a flow can cover many decades and the orientations of the particles are completely random with respect to the imaging setup. Learnable modules such as spatial transformer networks have been developed in the last decade to solve these problems [33]. We show in this study that their use can be avoided in the case of IPI by simply converting the speckle patterns to polar coordinates. After this initial conversion, a very classical CNN architecture can be used to perform particle shape recognition directly from the interferometric images. The next step should be the test of such a network for the analysis of particles whose interferometric images overlap [34]. These images contain a combination of two signals (due to the two particles) with an additional high-frequency modulation linked to the separation between the particles. The quantitative analysis of such patterns represents an important challenge.

The present study was carried out with both centrosymmetric and non-centrosymmetric particle shapes. This study, which used a CNN, showed 100% recognition in both cases. For comparison, in IPI, direct reconstructions of particle shapes can be performed from

interferometric images using phase retrieval algorithms. These methods are based on the equivalence existing between the 2D Fourier transform of the interferometric pattern and the 2D autocorrelation of the particle's shape. However, they present limitations. For example, reconstructions using the error reduction algorithm have been shown to be less accurate in the case of non-centrosymmetric particles due to the twin image problem [24]. The reconstruction of a Y-shaped particle can lead to a particle with six branches by the superposition of a Y-shaped particle with its symmetric twin image. In the same way, the reconstruction of an L-like particle can lead to a cross-like particle by the superposition of an L-shaped particle with its symmetric twin image. With the CNN, a Y-shaped particle (see Figure 1f) is never confused with a dendrite-like particle with six branches (see Figure 1c). In the same way, an L-like particle (see Figure 1d) is never confused with a cross-like particle (see Figure 1b). The CNN can thus be a very powerful tool to complete analyses carried out using algorithms of reconstruction or to replace them. It could actually be very powerful to combine both procedures through the development of a network inspired by physics-informed neural networks in radiative transfer and fluid mechanics [35,36].

The results that we have presented in this study are experimental results that contain different sources of noise. The creation of our experimental database required a few hours. During this time, different sources of noise were present: dust deposits on the optics and slight variations in the laser intensity and the optical alignment. In addition, we realized our database by randomly programming particles of different shapes and sizes with random numbers of micromirrors. For example, the smallest stick was composed of only 100 on-state micromirrors, while the biggest dendrite was composed of 17,200 on-state micromirrors. For all shapes of particles, the intensity of the interferometric images could thus vary very significantly. Nevertheless, particle classification from the interferometric images was possible with the CNN. In the future, a complete analysis of the influence of noise will require performing IPI in real conditions, i.e., with real rough particles and illumination with a pulsed laser.

In the present study, the database could be limited to 500 images per particle, which gave convincing results and a 100% recognition rate. In the future, it should be necessary to create a larger database in the case of real particles with a wider number of morphologies. In this case, the complexity of the CNN will have to be adapted and optimized. It will always be necessary to find the best compromise between the time to create the experimental database, computing time, memory capacity, and the quality of the results without overtraining.

## 5. Conclusions

In summary, a convolutional neural network was tested successfully to recognize the shapes of rough particles from their interferometric images. A conversion of the images to polar coordinates enabled size recognition despite the random orientations of the particles. To confirm the potentiality of the network, centrosymmetric and non-centrosymmetric particles were considered. In the case of non-centrosymmetric particles, shape recognition was possible despite the twin image problem that exists in this case. Shape recognition was not limited by the size of the particles. The CNN was validated for particle sizes that randomly cover a decade. Such a network should be very powerful in particle tomography experiments where multi-view systems are used and the quality of the 3D reconstruction is directly linked to the quality of the reconstruction of each individual view [37].

**Author Contributions:** Investigation, A.A., B.D., M.T. and M.B.; software, A.A., A.F. and Q.F.; writing—review and editing, A.A. and M.B.; supervision, M.B. All authors have read and agreed to the published version of the manuscript.

**Funding:** This research received no external funding.

**Institutional Review Board Statement:** Not applicable.

**Informed Consent Statement:** Not applicable.

**Data Availability Statement:** Data will be made available on reasonable request.

**Acknowledgments:** The authors acknowledge technical support from CORIA.

**Conflicts of Interest:** The authors declare no conflict of interest.

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
