# Peer review of "Particle Shape Recognition with Interferometric Particle Imaging Using a Convolutional Neural Network in Polar Coordinates"

_photonics, doi:10.3390/photonics10070779_

Round 1
Reviewer 1 Report
This manuscript by Alexis Abad deals with particle shape assessment from IPI pattern in polar coordinates with a CNN algorithm. As an application of CNN, the idea is generally elaborative and of practical use for irregular particle measurement. The methodology and results are overall clearly presented. This work can be acceptable for Photonics, after a revision addressing the following issues.
1) In lines 57-59, the author describes the particle interference results simulated by DMD as "look like" and "similar" to the real interference results, I think more specific instructions should be added to illustrate the degree of similarity between them, so as to better explain the practical application value of the method.
2) On page 4, the subfigure (c) mentioned in line 115 is not shown in Figure 4, a more careful check throughout the manuscript is suggested.
3) On page 5, Figure 5, I think it will be clearer if the legends were used to illustrate the different layers of the CNN model in Figure 5.
4) The author transforms the image from Cartesian to polar coordinates to deal with the poor rotational invariant ability of CNN. However, to our best knowledge, some modules have been proposed to improve the rotational invariant ability of CNN. Suggest checking the related work available at https://arxiv.org/abs/1506.02025.
5) The writing is generally fluent and understandable for me, a non-native English speaker, yet typo/grammar errors should be corrected, e.g., line 8 “the shape of rough particles”, line 155, “1.10-4”
6) A reword of the abstract is suggested.
As comments above.
Reviewer 2 Report
This paper proposes a new method to recognize the particle shape from their interferometric images. A conversion of the interferometric images in polar coordinates enables particle shape recognition. The paper is carefully done, and the results could be of great interest for shape classification in IPI. However, I have the following concerns before it can be accepted and published:
1.My main concern is that this paper lacks the comparison between shape classification results in cartesian coordinates and polar coordinates. Though confusion matrix in Fig.10 is ideal, the shape classification results in cartesian coordinates should be discussed.
2.Six different particle shapes are used in classification, as is shown in Fig.1. In Page 5, Line 134, this paper says that 500 speckle patterns are recorded for each particle’s shape, but how many separate shapes are utilized in the generation of speckle dataset for each particle? In my opinion this point should be clarified to show the applicability of trained network.
3.In Page 3, Line 85, why choose interferometric images with a size of 256×256 for classification? In other words, do the size of the cropped interferometric images effect the classification accuracy?
4.The configuration of IPI system is not mentioned in this paper. Is the system setup same with the IPI system in Ref.32?
Round 2
Reviewer 2 Report
The authors have answered the comments. I recommend the manuscript will be accepted.
Author Response
This is actually our answer to reviewer 3 (report received later), not to reviewer 2.
